# (5-Hydroxy-4-oxo-2-styryl-4*H*-pyridin-1-yl)-acetic Acid Derivatives as Multifunctional Aldose Reductase Inhibitors

**DOI:** 10.3390/molecules25215135

**Published:** 2020-11-04

**Authors:** Huan Chen, Xin Zhang, Xiaonan Zhang, Wenchao Liu, Yanqi Lei, Changjin Zhu, Bing Ma

**Affiliations:** School of Chemistry and Chemical Engineering, Beijing Institute of Technology, Beijing 100081, China; ch13264138964@163.com (H.C.); zx15811458104@163.com (X.Z.); zhangxiaonan965@163.com (X.Z.); liuwhhh@163.com (W.L.); 15313252939@163.com (Y.L.)

**Keywords:** aldose reductase inhibitors, antioxidant activity, pyridinones derivatives, molecular docking

## Abstract

As rate-limited enzyme of polyol pathway, aldose reductase (ALR2) is one of the key inhibitory targets for alleviating diabetic complications. To reduce the toxic side effects of the inhibitors and to decrease the level of oxidative stress, the inhibitory selectivity towards ALR2 against detoxicating aldehyde reductase (ALR1) and antioxidant activity are included in the design of multifunctional ALR2 inhibitors. Hydroxypyridinone derivatives were designed, synthesized and evaluated their inhibitory behavior and antioxidant activity. Notably, {2-[2-(3,4-dihydroxy-phenyl)-vinyl]-5-hydroxy-4-oxo-4*H*-pyridin-1-yl}-acetic acid (**7l**) was the most potent, with IC_50_ values of 0.789 μM. Moreover, **7l** showed excellent selectivity towards ALR2 with selectivity index 25.23, which was much higher than that of eparlestat (17.37), the positive control. More significantly, **7l** performed powerful antioxidative action. At a concentration of 1 μM, phenolic compounds **7l** scavenged DPPH radical with an inhibitory rate of 41.48%, which was much higher than that of the well-known antioxidant Trolox, at 11.89%. Besides, **7l** remarkably suppressed lipid peroxidation with a rate of 88.76% at a concentration of 100 μM. The binding mode derived from molecular docking proved that the derivatives were tightly bound to the activate site, suggesting strongly inhibitory action of derivatives against ALR2. Therefore, these results provided an achievement of multifunctional ALR2 inhibitors capable with potency for both selective ALR2 inhibition and as antioxidants.

## 1. Introduction

Diabetes mellitus (DM) is a metabolic disorder resulting from defects in insulin secretion, insulin action, or both. Accompanied by injury, dysfunction and organ failure, diabetic complications, which induced by long-term hypoglycaemia, are recognized as the main causes of morbidity and mortality in diabetic patients [1,2]. In the polyol pathway, large amounts of evidence proved that the overproduction of sorbitol and increased level of oxidative stress are the crucial reasons accelerating process of diabetic complications [3,4,5].

As the first and rate-determining enzyme in the polyol pathway, aldose reductase (ALR2, EC 1.1.1.21) reduces glucose into sorbitol in the presence of nicotinamide adenine dinucleotide phosphate (NADPH) [6,7,8,9]. (Figure 1) Subsequently, the NAD^+^-dependent sorbitol dehydrogenase oxidizes the intermediate sorbitol to fructose. Under normal condition, most of the cellular glucose is converted into glucose 6-phosphate and metabolized by hexokinase in glycolytic pathway [10,11]. Under hypoglycemia, however, due to the saturation of hexokinase, excess glucose is then metabolized by the polyol pathway, leading to accumulation of sorbitol and depletion in NADPH and NAD^+^ [9]. Since the high polarity of sorbitol blocks easy penetration through membranes and subsequent removal from tissues by diffusion, the accumulation of sorbitol eventually leads to osmotic imbalance, cell swelling, and membrane permeability changes [2]. Simultaneously, the disturbance of NADPH/NADP^+^ and NAD^+^/NADH ratios alter cellular redox potentials, resulting in the onset of hyperglycemic oxidative stress caused by both increased accumulation of reactive oxygen species (ROS) and weakened antioxidant defense [12]. In addition, associated with the elevation of polyol pathway, the increased level of fructose also accelerates the intracellular formation of advanced glycation end products and facilitates the further generation of ROS [13]. Although greatly increased levels of ROS lead to significant cellular damage when the antioxidant defenses are overwhelmed, the antioxidants may decrease the ROS concentration, and lower the oxidative stress of chronic diseases [14,15]. Therefore, to eliminate the abnormal accumulation of sorbitol and to alleviate the injury related to oxidative stress, the ALR2 inhibition and further reduction in the polyol pathway of glucose metabolism are attractive methods able to reverse or retard the progression of diabetic complications [16]. Due to pharmacokinetic drawbacks, adverse side effects or low efficacy, however, a number of ALR2 inhibitors (ARIs) in class of carboxylic acid derivatives have failed. Poor selectivity over detoxicating aldehyde reductase (ALR1, EC 1.1.1.2), which specifically metabolizing toxic aldehydes, is considered as one of the critical reasons for the side effects [17,18]. Therefore, selective inhibition of ALR2 is desirable in the design of new ARIs. Although ARL2 differential inhibitors were promising to selectively inhibit the reduction of hemiacetal aldose rather than that of toxic aldehyde [19], regarding its significant role in the detoxification of ALR1, selective ALR2 inhibition was still the major topic focused on ARI research [20,21,22,23].

Endowed with potent ALR2 inhibitory activity, compounds with carboxylic acid moiety were mostly developed among synthetic ARIs. (4-Oxo-2-thioxothiazolidin-3-yl) acetic acid [20], rhodanine-3-hippuric acid derivatives [21], 2-phenoxypyrido[3,2-b]pyrazin-3(4H)-one [22] and 2-(3-oxo-2*H*-[1,2,4]triazino[5,6-b]indol-5(3H)-yl) acetic acid [23] showed high inhibitory efficacy with attractive IC_50_ values (23~42 nM), suggesting that the carboxylic group is the key moiety preforming outstanding activity. However, the antioxidant properties of these inhibitors need to be improved.

Served as antivirus [24,25], anti-Alzheimer’s disease [26] and even anti-diabetes [27,28,29,30] agents, synthetic 3-hydroxy-4(1H)-pyridinone derivatives may have potential for the development of ARI. Interestingly, a large amount of evidence has proven that synthetic 3-hydroxy-4(1H)-pyridinone derivatives exhibited remarkable activity of antioxidant, and they ameliorated oxidative stress by directly neutralizing superoxide anions [31,32,33]. Notably, novel 2-styryl-1*H*-pyridin-4-ones showed significant radical scavenging activity [34], illustrating that these compounds may be the desirable candidates with antioxidant capacity fighting against oxidative stress. Thus, combining ALR2 inhibition with antioxidant action, hydroxyl pyridinones is favorable to be designed as antioxidative core of carbonxylic multifunctional ARI. Unfortunately, the inhibitory action of hydroxyl pyridinones against ALR2 remained undiscovered. In the present study, a new series of (5-hydroxy-4-oxo-2-styryl-4*H*-pyridin-1-yl)-acetic acid derivatives were designed to verify their ALR2 inhibitory actions and antioxidant behaviors.

## 2. Results and Discussion

### 2.1. Chemistry

The synthetic route of the (5-hydroxy-4-oxo-2-styryl-4*H*-pyridin-1-yl)-acetic acid derivatives (**7**) is outlined in Scheme 1. From the starting material of commercial available kojic acid, critical intermediate **3** was prepared by benzyl protection and followed chlorination. Different styryl side chains were readily linked to the C2-position of pyranones, to obtain 5-benzyloxy-2-styryl-pyran-4-ones (**5**) by Wittig reaction. Intermediates **6** with carboxylic group were obtained by the substitution of glycine, followed by the deprotection of benzyl group by hydrochloric acid to afford the target compounds **7a**–**i**. In addition, the intermediates **6d**,**e** were reacted with BBr_3_ to yield the desirable phenolic products **7k**–**l**.

The synthetic procedure preparing intermediate **5** from kojic acid was referenced by reported method [34]. Due to the poor solution in organic solvent, glycine was inactive in the substitution. To increase reactiveness of glycine, the optimization of reaction conditions preparing **6** was performed by reacting different equimolar quantities of intermediate **5** and glycine, in the presence of solvents of variable water/*i*-PrOH proportions. The optimum conditions were finally found by refluxing the reaction mixture with a high ratio of glycine as reactant and high ratio of water as solvent in the presence of NaOH. The structures of novel pyridinone-based carboxylic derivatives were confirmed by using spectral data ^1^H NMR, ^13^C NMR and HRMS. Among the title compounds, **7d** was taken as a representative compound for spectroscopic discussion. In the ^1^H NMR spectrum of newly synthesized compound **7d**, a singlet was observed at *δ* 5.10 (s, 2H) corresponding to CH_2_ located in N-substituted acetic acid, proving that carbonxylic group was introduced into pyridinone core. Coincident with result of ^1^H NMR, signal in ^13^C NMR of **7d** was detected at *δ* 160.71 corresponding to C=O of carbonxylic group. In addition, the compound was confirmed by HRMS in negative mode. All other derivatives exhibit identical results.

### 2.2. Enzyme Inhibition

All the title compounds were tested for their inhibitory activity against human recombinant ALR2. In order to investigate the selectivity toward to ALR2 against ALR1, only derivatives active in ALR2 inhibition were then subjected to evaluate for their inhibitory effect on human recombinant ALR1. The results of inhibitory activity against the enzymes were summarized in Table 1.

The results demonstrated that most of 5-hydroxy-2-styryl-1*H*-pyridin-4-ones derivatives showed good to excellent inhibitory activity against ALR2 with IC_50_ values varying from 0.789 to 17.11 μM. The structure-activity relationship revealed that the modification of some substituents on aromatic ring located in 2-styryl side chain could dramatically (*p* < 0.05) alter the inhibitory activity (**7c**, **7g**, **7i** and **7k**–**l**). Notably, {2-[2-(3,4-dihydroxy-phenyl)-vinyl]-5-hydroxy-4-oxo-4*H*-pyridin-1-yl}-acetic acid (**7l**) exhibited the most potent inhibitory effect among the tested analogs (IC_50_ = 0.789 μM). In addition, compound **7k** embodied with the C2-vinylphenolic group had much better inhibitory activity than that with C2-vinyl-methoxy-phenylic one (**7d**), demonstrating that the introduction of phenolic group enhanced the activity significantly (*p* < 0.05). Interestingly, the compounds with electronic-withdrawing groups at the phenyl group of the C2-styryl residues were less effective in the ALR2 inhibition rather than that with electronic-donating groups. Indeed, substituted with methanesulfonyl group at aromatic ring of the side chain (**7i**), the inhibitory activity decreased greatly (*p* < 0.05) compared to that of **7a**. However, the inhibitory activity was strengthened due to the increasing hydrophobicity of the alkyl group at C2-styryl (**7c** > **7b** > **7a**). In this case, the IC_50_ value of epalrestat, which served as a positive ARI, was comparable to the one reported under similar experimental conditions [35], ensuring the validity of the tests.

To estimate the selectivity, the title compounds which presented preferable activity of ALR2 inhibition (IC_50_ ≤ 5 μM), as well as compound **7a** for comparison, were tested for the inhibition against ALR1. All of the tested compounds were found to be slightly active with inhibition percentage in 100 μM, demonstrating their selectivity for ALR2. Compound **7l** which exhibited strongest inhibition of ALR2 had an excellent selectivity with the selectivity index (SI) of 25.23, which was much higher than that of epalrestat.

### 2.3. Antioxidant Activity

#### 2.3.1. DPPH Radical Scavenging Activity

Oxidative stress is one of crucial factors in the pathological processes of many disorders, including chronic diabetic complications. To retard the progression of oxidative stress and to ameliorate the disease, ARIs are expected to have significant antioxidant properties. Thus, the antioxidant capacities of the compounds which were effective in the ALR2 inhibition were estimated.

In this part, the antioxidant activity was measured by an intrinsic chemical reactivity toward radicals, in a homogeneous reaction system of the methanol solution of 1,1-diphenyl-2-trinitrophenylhydrazine (DPPH) (0.2 mM), and 6-hydroxy-2,5,7,8-tetramethyl- chroman-2-carboxylic acid (Trolox) served as a reference compound. The derivatives **7c**, **7g** and **7k**–**l** which already had exhibited good selectivity toward to ALR2 inhibition were tested for DPPH radical scavenging activity, as well as compound **7a** serving as comparison. Interestingly, much superior to well-known antioxidant Trolox and reported deferiprone (3-hydroxy-1,2-dimethyl-1*H*-pyridin-4-one) [36], phenolic derivatives **7k**–**l** showed excellent DPPH radical scavenging ability, revealing that the phenolic substituents might be a key structure enhancing DPPH free radical scavenging activity (Table 2). Particularly, at a concentration of 1 μM, compound **7l** had most favorable antioxidant properties, achieving 41.48% of inhibition rate. In addition, all the compounds without hydroxyl subunit at C2-styryl exhibited moderate antioxidant activity, proving that 5-hydroxy-2-styryl-1*H*-pyridin-4-ones were qualified antioxidants.

#### 2.3.2. Lipid Peroxidation Suppression

Lipid peroxidation reactions involve free radical attacks on unsaturated fatty acid, producing free radicals and lipid peroxides which eventually harm the organism and usually decompose to form poisonous malonaldehyde (MDA). Thiobarbituric acid reactive substances (TBARS), which generated from the condensation of thiobarbituric acid and malondialdehyde, are considered as key biomarkers to determine the level of lipid peroxidation. For further identification that the derivatives tested in DPPH scavenging assay can be effective in heterogeneous oxidative pathway, the suppression effect of compounds on the hydroxyl radical-dependent lipoperoxidation in rat brain homogenate oxidant system which induced by Fe(III)/ascorbic acid was also evaluated.

Compounds **7a**, **7c**, **7g** and **7k**–**l** that were already described in the DPPH radical scavenging activity were investigated in the suppression of lipid peroxidation (Table 3). Consistent with results from the DPPH radical scavenging assay, the lipid peroxidation suppression showed that all tested compounds were active in the indicative heterogeneous assay, which was comparable to reported antioxidant deferiprone [37]. Remarkably, greater than the positive control Trolox and deferiprone, compounds **7k**–**l** showed an attractive activity inhibiting the production of MDA, further confirming that the hydroxyl styryl structures of (5-hydroxy-4-oxo-2-styryl-4*H*-pyridin-1-yl)-acetic acid derivatives were important in the antioxidant activity.

Combined with the conclusions from both DPPH radical scavenging assay and the lipid peroxidation suppression experiment, 5-hydroxy-4-oxo-2-styryl-4*H*-pyridine derivatives have been proved their good antioxidant capacity. Compared favorably with Trolox, the typical antioxidant agent, hydroxystyryl derivatives provided excellent anti-oxidative activity, revealing that the hydroxyl groups in C2- styryl side chain were critical structures behaving like the ideal antioxidant.

### 2.4. Molecular Docking

To propose the binding mode of 5-hydroxy-4-oxo-2-styryl-4*H*-pyridine derivatives at a molecular level and to further illustrate the relationship between structure and ALR2 inhibitory activity, molecular docking assays of compounds which endowed with good activity in the ALR2 inhibition were performed by software Molegro Virtual Docker, version 6.0. As shown in Figure 2, compounds **7c**, **7d**, **7g** and **7k**–**l**, as well as compound **7a** and **7i** which set as comparison, were docked well into the active site of the human ALR2/NADP+/lidorestat complex (PDB code: 1Z3N) [38]. By strong hydrogen bonds interaction and pi-pi stacking with the side chain of the residues, the carboxyl groups of these compounds were deeply inserted into the anion binding pocket consisted by His 110, Trp 111, Tyr 48, Val 47, Trp 20 and cofactor NADPH. However, the carboxyl group of compound **7i** partially “flowed out” of the anion binding pocket, confirming that **7i** was inactive in inhibiting ALR2, according to the minimum requirement of a qualified inhibitor posed by Costantino et al. [39]. Interestingly, hydrogen bonds were created between the carboxylic groups and His110/Trp 111 residues, as well as between hydroxyl groups in pyridinone core and Val 47/Trp 20 residues, which lying opposite side of His 110 and Trp 111 residues in this pocket. Thus, it is suggested that these carboxylic pyridinones might readily and totally blockade the activate site catalyzing the reduction of aldose. Notably, an additional hydrogen bond was formed vigorously between the hydroxyl group in C2-styryl side chain of **7l** and residue Thr 103 which located in specificity pocket, letting the specificity pocket be closed, which is in line with the recently published docking result of potent inhibitors, cemtirestat O-analog [23]. Thus, it was proved that **7l** was much more active than the others.

To further understand the inhibitory action of 5-hydroxy-4-oxo-2-styryl-4*H*-pyridine derivatives in detail, the binding energy and inhibitory constant (Ki) for **7a**, **7c**, **7d**, **7g** and **7k**–**l** were calculated (Table 4). Since poor interaction with enzyme complex, data of **7i** was out of consideration. In general, ranging from 2.06 to 3.23 μM, the predicted Ki data of all compounds were closely related to the IC_50_ values, verifying the ALR2 inhibitory actions of the derivatives. Significantly, lower Ki for **7c**, **7d**, **7g** and **7k**–**l** were found rather than that for **7a**, demonstrating better ALR2 affinities of the derivatives substituted with electronic-donating groups or the alkyl groups at C2-styryl side chains. This result was entirely consistent with the result from ALR2 inhibitory assay, of which the IC_50_ data of **7c**, **7d**, **7g** and **7k**–**l** were lower than that of **7a**. Notably, among all of these compounds, **7l,** which acquired most favorable IC_50_ 0.789 μM, had the minimum value of Ki 2.06 μM, illustrating that phenolic **7l** was prominently potent when binding into the ALR2 active site. Tightly interacting with anion binding site and provided with lowest value of Ki, phenolic **7l** thus had the most desirable binding affinity, along with the pattern of ALR2 inhibition.

To investigate the selectiveness towards ALR2 over ALR1, the binding behaviors of derivatives with the ALR1-NADP+ complex (PDB code: 1ZUA) [40] were also discussed. The binding energies of compounds **7a**, **7c** and **7k**–**l** were calculated and displayed in Table 4. Remarkably, in agreement with the results in the inhibition experiment against ALR1, these data were explicit that **7l** exhibited a higher reranked score than **7a** did, proving that this phenolic compound acquired good selectivity for ALR2.

## 3. Materials and Methods

### 3.1. General

Melting points were recorded on an X-4 microscopic melting point apparatus (Shanghai Jingke Instrument Co., Ltd, Shanghai, China) and are uncorrected. All reactions were routinely checked by TLC on silica gel Merck 60F254. The NMR spectra were recorded on a Bruker Ascend 400 M spectrometer (400 MHz for ^1^H NMR, 100 MHz for ^13^C NMR, Bruker Corporation, Billerica, MA, USA) and chemical shifts were given in *δ* units (ppm) relative to internal standard TMS and refer to DMSO-*d6* solutions. HRMS (ESI) was performed using an AGILENT LC/MS (Agilent Technologies Inc., Palo Alto, CA, USA). Analysis of sample purity was performed on a Hitachi D-2000 Elite HPLC system (Hitachi, Ltd., Tokyo, Japan). HPLC conditions were the following: Inertsil ODS-2 250 mm × 10 mm, 5 mm column; mobile phase: CH_3_CN (0.1% TFA) /CH_3_OH = 1/1, for 10 min; room temperature; flow rate: 1 mL min^−1^; detection at *λ* 254 nm. All final compounds in biological assays have a purity of ≥95%.

### 3.2. Synthetic Procedures Preparing Title Compounds

Preparation of compounds **2**–**4** was reported previously [34]. With partial modification, the procedures for preparing title compounds are as followed.

#### 3.2.1. Synthetic Procedure for 5-benzyloxy-2-hydroxymethyl-pyran-4-one (**2**)

To a solution of kojic acid (**1**) (28.4 g, 0.2 mol) in isopropanol (200 mL) sodium hydroxide (8.8 g, 0.2 mol) aqueous solution (20 mL) was added, and the mixture was heated to reflux. Benzyl chloride (28 g, 0.2 mol) was added dropwise into the mixture over 30 min, and the resulting mixture was refluxed for 4 h. After removal of the solvent by rotary evaporation, the brown solid was washed with water (80 mL) followed by acetate ester (200 mL) then recrystallized from ethanol to give the pure product (32.28 g, 69.6%). ^1^H NMR (400 MHz, DMSO) *δ* 7.91 (s, 1H), 7.19 (m, 5H), 6.17 (s, 1H), 4.81 (s, 2H), 4.14 (s, 2H).

#### 3.2.2. Synthetic Procedure for 5-benzyloxy-2-chloromethyl-pyran-4-one (**3**)

**2** (46.4 g, 0.2 mol) was added in flask containing 300 mL dichloromethane, 22 mL thionyl chloride was added dropwise at 0 °C. The reaction mixture was stirred at room temperature for 2 h. After filtration, the solid was washed with petroleum 200 mL to get white solid (55.66 g, 74%). ^1^H NMR (400 MHz, DMSO) *δ* 7.53 (s, 1H), 7.32 (m, 5H), 6.35 (s, 1H), 5.16 (s, 2H), 4.22 (s, 2H).

#### 3.2.3. Synthetic Procedure for ((5-(benzyloxy)-4-oxo-4*H*-pyran-2-yl)methyl)triphenylphosphonium chloride (**4**)

**3** (25 g, 0.1 mol) was dissolved in dried tetrahydrofuran (100 mL), triphenylphosphine (26.2 g, 0.1 mol) was added, and the mixture was refluxed overnight. The solvent was then evaporated, and the residue washed with ethyl acetate to get the solid (22.12 g, 43.2%).

#### 3.2.4. General Procedure Preparing 5-benzyloxy-2-styryl-pyran-4-one (**5**)

Into the mixture of **4** (5.13 g, 10 mmol) and benzaldehyde (11 mmol) in dichloromethane (20 mL), aqueous solution of sodium hydroxide (50%, 6 mL) was dropwise added over 15 min, and the mixture was stirred at room temperature for 2 h. Water was added and the resulting solution was extracted with dichloromethane (50 mL × 3). The combined organic layers were gathered and concentrated in vacuo, and the residue was then recrystallized from ethanol to get a white or yellow solid (25~87%).

*5-Benzyloxy-2-styryl-pyran-4-one* (**5a**) (78%). ^1^H NMR (400 MHz, CDCl_3_) *δ* 7.55 (s, 1H), 7.49 (d, *J* = 7.1 Hz, 2H), 7.31 (m, 9H), 6.63 (d, *J* = 16.1 Hz, 1H), 6.39 (s, 1H), 5.11 (s, 2H). ^13^C NMR (100 MHz, *d*6-DMSO) *δ* 174.41, 161.44, 146.37, 139.61, 137.02, 135.46, 135.28, 130.04, 129.41, 128.42, 128.37, 128.34, 128.00, 120.63, 112.58, 59.50.

*5-Benzyloxy-2-(2-p-tolyl-vinyl)-pyran-4-one* (**5b**) (71%). ^1^H NMR (400 MHz, CDCl_3_) *δ* 7.58 (s, 1H), 7.67 (d, *J* = 7.2 Hz, 2H), 7.30 (m, 8H), 6.54 (d, *J* = 16.0 Hz, 1H), 6.30 (s, 1H), 5.13 (s, 2H), 2.32 (s, 3H). ^13^C NMR (100 MHz, *d*6-DMSO) *δ* 174.93, 161.25, 146.21, 139.17, 137.64, 135.77, 135.79, 129.64, 129.51, 128.65, 128.38 128.31, 128.12 120.79, 112.45, 58.41, 21.47.

*5-Benzyloxy-2-[2-(4-tert-butyl-phenyl)-vinyl]-pyran-4-one* (**5c**) (46%). ^1^H NMR (400 MHz, CDCl_3_) *δ* 7.55 (s, 1H), 7.60 (d, *J* = 7.2 Hz, 2H), 7.32 (m, 8H), 6.61 (d, *J* = 16.0 Hz, 1H), 6.44 (s, 1H), 5.15 (s, 2H), 1.41 (s, 9H). ^13^C NMR (100 MHz, *d*6-DMSO) *δ* 174.35, 161.63, 146.01, 139.91, 137.87, 135.45, 134.91, 129.98, 128.79, 128.88, 128.36, 128.33, 127.83, 120.12, 112.27, 56.84, 34.95, 31.45.

*5-Benzyloxy-2-[2-(4-methoxy-phenyl)-vinyl]-pyran-4-one* (**5d**) (44%). ^1^H NMR (400 MHz, CDCl_3_) *δ* 7.53 (s, 1H), 7.38 (m, 7H), 7.27 (d, *J* = 16.0 Hz, 1H), 6.91 (m, 2H), 6.50 (d, *J* = 16.0 Hz, 1H), 6.37 (s, 1H), 5.12 (s, 2H), 3.89 (s, 3H). ^13^C NMR (100 MHz, *d*6-DMSO) *δ* 173.01, 160.88, 146.18, 139.27, 136.94, 136.03, 135.88, 135.41, 130.14, 128.43, 128.39, 128.21, 127.12, 127.03, 113.80, 74.32, 56.55.

*5-Benzyloxy-2-[2-(3,4-dimethoxy-phenyl)-vinyl]-pyran-4-one* (**5e**) (48%). ^1^H NMR (400 MHz, CDCl_3_) *δ* 7.55 (s, 1H), 7.36 (m, 5H), 7.28 (d, *J* = 4.7 Hz, 1H), 6.64 (m, 3H), 6.48 (t, *J* = 2.5 Hz, 1H), 6.36 (s, 1H), 5.12 (s, 2H), 3.83 (s, 6H). ^13^C NMR (100 MHz, *d*6-DMSO) *δ* 174.40, 161.22, 147.51, 139.49, 137.57, 137.01, 135.28, 129.41, 128.42, 128.37, 127.31, 121.48, 120.63, 112.58, 106.34, 102.05, 59.33.

*5-Benzyloxy-2-[2-(4-fluoro-phenyl)-vinyl]-pyran-4-one* (**5f**) (56%). ^1^H NMR (400 MHz, CDCl_3_) *δ* 7.55 (s, 1H), 7.30 (m, 8H), 7.21 (d, *J* = 9.7 Hz, 1H), 7.11 (dd, *J* = 8.2, 2.4 Hz, 1H), 6.63 (d, *J* = 16.1 Hz, 1H), 6.41 (s, 1H), 5.12 (s, 2H). ^13^C NMR (100 MHz, *d*6-DMSO) *δ* 174.42, 163.09, 161.36, 146.58, 139.52, 137.49, 136.03, 133.88, 131.41, 128.39, 128.21, 127.12, 124.46, 122.16, 116.59, 114.07, 112.91, 57.41.

*5-Benzyloxy-2-[2-(3,4-difluoro-phenyl)-vinyl]-pyran-4-one* (**5g**) (63%). ^1^H NMR (400 MHz, CDCl_3_) *δ* 7.55 (s, 1H), 7.21 (m, 10H), 6.64 (d, *J* = 16.1 Hz, 1H), 6.39 (s, 1H), 5.10 (s, 2H). ^13^C NMR (100 MHz, *d*6-DMSO) *δ* 176.12, 162.77, 146.92, 144.28, 135.46, 130.69, 128.42, 128.37, 128.34, 125.38, 123.68, 120.79, 111.07, 108.86, 104.18, 59.74.

*5-Benzyloxy-2-[2-(4-trifluoromethyl-phenyl)-vinyl]-pyran-4-one* (**5h**) (66%). ^1^H NMR (400 MHz, CDCl_3_) *δ* 7.55 (s, 1H), 7.28 (m, 8H), 7.26 (d, *J* = 9.3 Hz, 1H), 7.08 (dd, *J* = 8.4, 2.1 Hz, 1H), 6.60 (d, *J* = 16.1 Hz, 1H), 6.46 (s, 1H), 5.15 (s, 2H). ^13^C NMR (100 MHz, *d*6-DMSO) *δ* 174.59, 160.28, 156.79, 146.82, 141.76, 139.23, 135.28, 132.84, 129.05, 128.63, 128.42, 128.37, 128.34, 124.60, 120.83, 113.28, 59.50.

*5-Benzyloxy-2-[2-(4-methanesulfonyl-phenyl)-vinyl]-pyran-4-one* (**5i**) (78%). ^1^H NMR (400 MHz, CDCl_3_) *δ* 7.55 (s, 1H), 7.31 (m, 8H), 7.20 (d, *J* = 9.5 Hz, 1H), 7.11 (dd, *J* = 8.0, 2.5 Hz, 1H), 6.83 (d, *J* = 16.1 Hz, 1H), 6.55 (s, 1H), 5.15 (s, 2H), 3.30 (s, 3H). ^13^C NMR (100 MHz, *d*6-DMSO) *δ* 173.03, 164.63, 149.25, 143.46, 137.77, 136.92, 134.94, 128.82, 128.73, 128.18 127.65, 127.25, 122.72, 120.63, 111.74, 58.32, 41.62.

*N-{4-[2-(5-Benzyloxy-4-oxo-4*H*-pyran-2-yl)-vinyl]-phenyl}-acetamide* (**5j**) (25%). ^1^H NMR (400 MHz, CDCl_3_) *δ* 10.23 (s, 1H), 8.22 (s, 1H), 7.63 (m, 4H), 7.40 (d, J = 2.6 Hz, 2H), 7.14 (m, 4H), 5.14 (s, 2H), 2.10 (s, 3H). ^13^C NMR (100 MHz, *d*6-DMSO) *δ* 174.41, 171.29, 162.64, 151.69, 146.10 139.04, 137.02, 136.08, 131.49, 129.88, 129.57, 128.42, 128.37, 128.34, 128.00, 112.58, 110.29, 59.50, 24.51.

#### 3.2.5. General Procedure Preparing (5-Benzyloxy-4-oxo-2-styryl-4*H*-pyridin-1-yl)-acetic Acid (**6**)

**5** (3 mmol) dissolved in isopropanol (5 mL) and the mixture was added into glycine (20 mmol) aqueous solution 15 mL, then followed by aqueous solution of sodium hydroxide (50%, 6 mL) added into the mixture to ensure the pH value maintain 8~9. The resulting suspension was refluxed for 16~36 h, till the mixture presented as clear solution. After evaporation to remove solvent isopropanol, 8 M HCl solution was added to adjust the pH to 2. A precipitate was obtained and filtered, and then washed by water to get white or yellow solid. (12~27%)

*(5-Benzyloxy-4-oxo-2-styryl-4*H*-pyridin-1-yl)-acetic acid* (**6a**) (22%). ^1^H NMR (400 MHz, *d*6-DMSO) *δ* 7.58 (m, 2H), 7.51–7.19 (m, 10H), 7.01 (m, 2H), 5.49 (s, 2H), 5.02 (s, 2H). ^13^C NMR (100 MHz, *d*6-DMSO) *δ* 169.31, 163.24, 160.83, 147.66, 139.92, 137.43, 135.94, 135.26, 131.82, 129.92, 128.46, 128.34, 128.28, 128.11, 121.87, 111.72, 59.94, 56.31.

*[5-Benzyloxy-4-oxo-2-(2-p-tolyl-vinyl)-4*H*-pyridin-1-yl]-acetic acid* (**6b**) (25%). ^1^H NMR (400 MHz, *d*6-DMSO) *δ* 7.57 (m, 2H), 7.57–7.22 (m, 10H), 7.06 (m, 2H), 5.44 (s, 2H), 5.01 (s, 2H), 2.33 (s, 3H). ^13^C NMR (100 MHz, *d*6-DMSO) *δ* 169.27, 162.24, 161.23, 146.64, 140.37, 137.13, 136.01, 135.21, 129.84, 129.42, 128.79, 128.48 128.33, 128.06, 122.43, 115.56, 58.84, 56.17, 22.42.

*{5-Benzyloxy-2-[2-(4-tert-butyl-phenyl)-vinyl]-4-oxo-4*H*-pyridin-1-yl}-acetic acid* (**6c**) (14%). ^1^H NMR (400 MHz, *d*6-DMSO) *δ* 7.58 (m, 2H), 7.52–7.13 (m, 10H), 6.98 (m, 2H), 5.40 (s, 2H), 5.06 (s, 2H), 1.31 (s, 9H). ^13^C NMR (100 MHz, *d*6-DMSO) *δ* 168.94, 163.60, 161.92, 145.91, 140.38, 136.87, 135.28, 135.41, 130.24, 129.33, 128.65, 128.42, 128.28, 127.66, 120.01, 114.68, 59.71, 55.91, 34.03, 30.33.

*{5-Benzyloxy-2-[2-(4-methoxy-phenyl)-vinyl]-4-oxo-4*H*-pyridin-1-yl}-acetic acid* (**6d**) (25%). ^1^H NMR (400 MHz, *d*6-DMSO) *δ* 7.62 (s, 1H), 7.33 (m, 12H), 5.32 (s, 2H), 5.01 (s, 2H), 3.86 (s, 3H). ^13^C NMR (100 MHz, *d*6-DMSO) *δ* 168.85, 161.41, 160.23, 147.31, 138.61, 137.45, 136.00, 135.71, 135.22, 129.63, 128.84, 128.71, 128.62, 127.51, 127.39, 115.06, 78.04, 56.92, 54.17.

*{5-Benzyloxy-2-[2-(3,4-dimethoxy-phenyl)-vinyl]-4-oxo-4*H*-pyridin-1-yl}-acetic acid* (**6e**) (13%). ^1^H NMR (400 MHz, *d*6-DMSO) *δ* 7.77 (s, 1H), 7.38 (m, 7H), 7.06 (d, *J* = 16.2 Hz, 1H), 6.75 (d, *J* = 2.0 Hz, 2H), 6.47 (s, 1H), 5.30 (s, 2H), 5.11 (s, 2H), 3.78 (s, 6H). ^13^C NMR (100 MHz, *d*6-DMSO) *δ* 169.04, 160.19, 159.41, 148.86, 140.83, 138.76, 136.69, 135.92, 130.53, 129.10, 128.91, 127.58, 122.33, 121.20, 113.44, 107.08, 101.45, 57.26, 55.23.

*{5-Benzyloxy-2-[2-(4-fluoro-phenyl)-vinyl]-4-oxo-4*H*-pyridin-1-yl}-acetic acid* (**6f**) (25%). ^1^H NMR (400 MHz, *d*6-DMSO) *δ* 7.80 (m, 12H), 7.12(s, 1H), 5.28 (s, 2H), 5.20 (s, 2H). ^13^C NMR (100 MHz, *d*6-DMSO) *δ* 169.18, 161.92, 161.64, 160.22, 140.43, 138.96, 137.31, 137.11, 132.93, 132.07, 129.22, 128.78, 127.99, 125.84, 122.18, 117.24, 113.63, 110.42, 59.68, 57.24.

*{5-Benzyloxy-2-[2-(3,4-difluoro-phenyl)-vinyl]-4-oxo-4*H*-pyridin-1-yl}-acetic acid* (**6g**) (20%). ^1^H NMR (400 MHz, *d*6-DMSO) *δ* 7.42 (s, 1H), 7.36 (m, 5H), 7.30 (s, 1H), 7.20 (m, 2H), 6.96 (s, 1H), 6.83 (d, *J* = 16.1 Hz, 1H), 6.70 (s, 1H), 5.33 (s, 2H), 5.12 (s, 2H). ^13^C NMR (100 MHz, *d*6-DMSO) *δ* 169.84, 160.63, 159.28, 147.05, 145.20, 135.83, 131.04, 129.74, 128.84, 128.34, 126.97, 123.41, 121.04, 110.26, 107.43, 101.87, 58.22, 56.81.

*{5-Benzyloxy-4-oxo-2-[2-(4-trifluoromethyl-phenyl)-vinyl]-4*H*-pyridin-1-yl}-acetic acid* (**6h**) (18%). ^1^H NMR (400 MHz, *d*6-DMSO) *δ* 8.24 (d, *J* = 8.7 Hz, 2H), 7.86 (d, *J* = 8.7 Hz, 2H), 7.52 (d, *J* = 7.1 Hz, 1H), 7.47 (d, *J* = 7.3 Hz, 2H), 7.40 (m, 3H), 7.34 (m, 2H), 6.88 (s, 1H), 5.21 (s, 2H), 5.16 (s, 2H). ^13^C NMR (100 MHz, *d*6-DMSO) *δ* 169.22, 160.83, 159.12, 155.67, 147.28, 140.66, 139.49, 136.02, 132.91, 128.97, 128.52, 128.32, 128.28, 127.33, 125.32, 121.70, 112.76, 59.84, 55.82.

*{5-Benzyloxy-2-[2-(4-methanesulfonyl-phenyl)-vinyl]-4-oxo-4*H*-pyridin-1-yl}-acetic acid* (**6i**) (27%). ^1^H NMR (400 MHz, *d*6-DMSO) *δ* 8.20 (d, *J* = 8.5 Hz, 2H), 7.88 (d, *J* = 8.3 Hz, 2H), 7.45 (d, *J* = 7.6 Hz, 1H), 7.33 (d, *J* = 7.1 Hz, 2H), 7.28 (m, 3H), 7.13 (m, 2H), 6.92 (s, 1H), 5.17 (s, 2H), 5.05 (s, 2H). ^13^C NMR (100 MHz, *d*6-DMSO) *δ* 169.63, 161.08, 159.81, 148.74, 144.40, 137.26, 136.48, 130.26, 129.42, 128.68, 128.55 127.71, 127.04, 123.58, 121.02, 111.98, 58.77, 56.11, 41.62.

*{2-[2-(4-Acetylamino-phenyl)-vinyl]-5-benzyloxy-4-oxo-4*H*-pyridin-1-yl}-acetic acid* (**6j**) (12%). ^1^H NMR (400 MHz, *d*6-DMSO) *δ* 10.21 (s, 1H), 8.26 (s, 1H), 7.66 (m, 4H), 7.43 (d, *J* = 2.6 Hz, 2H), 7.19 (m, 4H), 5.29 (s, 2H), 5.12 (s, 2H), 2.08 (s, 3H). ^13^C NMR (100 MHz, *d*6-DMSO) *δ* 169.59, 161.28, 160.02, 158.32, 154.32, 147.48 138.62, 137.88, 136.01, 131.26, 129.54, 129.21, 128.44, 128.40, 128.21, 128.13, 113.63, 111.74, 59.50, 56.65, 24.51.

#### 3.2.6. General Procedure Preparing (5-hydroxy-4-oxo-2-styryl-4*H*-pyridin-1-yl)-acetic Acid (**7a–l**)

Method A: **6** (2 mmol) was dissolved in hydrochloric acid (6 N, 10 mL). The mixture was refluxed for 24 h, resulting solid was filtered and recrystallized from ethanol to yield compounds **7a–i** with white to yellow solid. (71~88%)

Method B: Boron tribromide (6 eq) was added dropwise at −20 °C under nitrogen to a solution of **6d** (or **6e**) (2 mmol) in dichlomethane (20 mL), then the resulting mixture was slowly warmed to room temperature and stirred overnight. Methanol (10 mL) was slowly added, and the mixture stirred for another 0.5 h. The resulting mixture was evaporated and recrystallized from methanol to get yellow solid **7k** (or **7l**).

Detailed characterizations and spectra of all the products are described in the Appendix A to this paper.

*(5-Hydroxy-4-oxo-2-styryl-4*H*-pyridin-1-yl)-acetic acid* (**7a**) (81%). Purity: 98.33%; m.p.: 254–255 °C; ^1^H NMR (400 MHz, *d*6-DMSO) *δ* 7.66 (s, 3H), 7.43 (m, 3H), 7.27 (d, *J* = 2.6 Hz, 1H), 7.20 (d, *J* = 2.8, 1H), 6.83 (s, 1H), 5.11 (s, 2H). ^13^C NMR (100 MHz, *d*6-DMSO) *δ* 169.47, 168.04, 146.77, 146.20, 136.01, 127.50, 119.30, 110.68, 55.10. HRMS (ESI) *m/z* calcd. for C_15_H_12_NO_4_^−^ [M − H]^−^ 270.0772, found 270.0786.

*[5-Hydroxy-4-oxo-2-(2-p-tolyl-vinyl)-4*H*-pyridin-1-yl]-acetic acid* (**7b**) (88%). Purity: 99.12%; m.p.: 256–257 °C; ^1^H NMR (400 MHz, *d*6-DMSO) *δ* 8.28 (s, 1H), 7.69 (s, 1H), 7.63 (d, *J* = 8.7 Hz, 2H), 7.44 (d, *J* = 16.1, 1H), 7.28 (m, 3H), 5.50 (s, 2H), 2.35 (s, 3H). ^13^C NMR (100 MHz, *d*6-DMSO) *δ* 168.44, 161.60, 144.80, 140.33, 139.96, 132.85, 129.98, 129.45, 116.89, 110.39, 56.64, 21.49. HRMS (ESI) *m/z* calcd. for C_16_H_14_NO_4_^−^ [M − H]^−^ 284.0928, found 284.0941.

*{2-[2-(4-tert-Butyl-phenyl)-vinyl]-5-hydroxy-4-oxo-4*H*-pyridin-1-yl}-acetic acid* (**7c**) (75%). Purity: 98.12%; m.p.: 241–242 °C; ^1^H NMR (400 MHz, *d*6-DMSO) *δ* 7.56 (m, 3H), 7.44 (d, *J* = 2.6 Hz, 2H), 7.20 (d, *J* = 8.9 Hz, 1H), 7.11 (d, *J* = 8.8 Hz, 1H), 6.69 (s, 1H), 5.03 (s, 2H), 1.30 (s, 9H). ^13^C NMR (100 MHz, *d*6-DMSO) *δ* 169.76, 169.74, 152.37, 146.14, 136.75, 133.46, 126.02, 118.86, 110.52, 54.62, 34.95, 31.45. HRMS (ESI) *m/z* calcd. for C_19_H_20_NO_4_^−^ [M − H]^−^ 326.1398, found 326.1405.

*{5-Hydroxy-2-[2-(4-methoxy-phenyl)-vinyl]-4-oxo-4*H*-pyridin-1-yl}-acetic acid* (**7d**) (71%). Purity: 96.53%; m.p.: 240–242 °C; ^1^H NMR (400 MHz, *d*6-DMSO) *δ* 7.63 (m, 3H), 7.27 (d, *J* = 16.2 Hz, 2H), 7.00 (m, 2H), 6.82 (s, 1H), 6.69 (s, 1H), 5.10 (s, 2H), 3.80 (s, 3H). ^13^C NMR (100 MHz, *d*6-DMSO) *δ* 169.50, 160.71, 146.61, 137.32, 129.64, 128.59, 116.70, 114.75, 110.16, 71.27, 55.77. HRMS (ESI) *m/z* calcd. for C_16_H_14_NO_5_^−^ [M − H]^−^ 300.0877, found 300.0874.

*{2-[2-(3,4-Dimethoxy-phenyl)-vinyl]-5-hydroxy-4-oxo-4*H*-pyridin-1-yl}-acetic acid* (**7e**) (80%). Purity: 95.84%; m.p.: 220–222 °C; ^1^H NMR (400 MHz, *d*6-DMSO) *δ* 7.71 (d, *J* = 2.2 Hz, 1H), 7.53 - 7.42 (m, 4H), 7.23 (d, *J* = 2.1 Hz, 2H), 6.66 (s, 1H), 5.18 (s, 2H), 3.89 (s, 3H), 3.84 (s, 3H). ^13^C NMR (100 MHz, *d*6-DMSO) *δ* 168.44, 163.00, 159.70, 149.41, 145.04, 135.86, 135.46, 131.42, 130.81, 116.74, 115.40, 106,76, 98.91, 71.96, 56.32, 56.04. HRMS (ESI) *m/z* calcd. for C_17_H_16_NO_6_^−^ [M − H]^−^ 330.0983, found 330.0979.

*{2-[2-(4-Fluoro-phenyl)-vinyl]-5-hydroxy-4-oxo-4*H*-pyridin-1-yl}-acetic acid* (**7f**) (87%). Purity: 99.19%; m.p.: 262–263 °C; ^1^H NMR (400 MHz, *d*6-DMSO) *δ* 7.71 (m, 2H), 7.49 (s, 1H), 7.25 (m, 3H), 7.12 (m, 1H), 6.61 (s, 1H), 4.99 (s, 2H). ^13^C NMR (100 MHz, *d*6-DMSO) *δ* 170.76, 169.91, 164.10, 161.65, 147.57, 145.80, 135.29, 130.01, 129.93, 125.17, 116.06, 110.62, 54.34. HRMS (ESI) *m/z* calcd. for C_15_H_11_FNO_4_^−^ [M − H]^−^ 288.0678, found 288.0677.

*{2-[2-(3,4-Difluoro-phenyl)-vinyl]-5-hydroxy-4-oxo-4*H*-pyridin-1-yl}-acetic acid* (**7g**) (74%). Purity: 98.57%; m.p.: 255–256 °C; ^1^H NMR (400 MHz, *d*6-DMSO) *δ* 7.86 (dd, *J* = 8.1, 2.2 Hz, 1H), 7.50 (s, 1H), 7.33 (t, *J* = 8.1 Hz, 1H), 7.16 (m, 3H), 6.58 (s, 1H), 4.95 (s, 2H). ^13^C NMR (100 MHz, *d*6-DMSO) *δ* 170.76, 169.83, 147.70, 145.54, 130.50, 125.21, 122.62, 120.71, 110.92, 104.92, 54.37. HRMS (ESI) *m/z* calcd. for C_15_H_10_F_2_NO_4_^−^ [M − H]^−^ 306.0583, found 306.0587.

*{5-Hydroxy-4-oxo-2-[2-(4-trifluoromethyl-phenyl)-vinyl]-4*H*-pyridin-1-yl}-acetic acid* (**7h**) (83%). Purity: 97.80%; m.p.: 267–268 °C; ^1^H NMR (400 MHz, *d*6-DMSO) *δ* 8.34 (s, 1H), 7.97 (d, *J* = 2.4 Hz, 2H), 7.84 (d, *J* = 2.2 Hz, 2H), 7.75 (s, 1H), 7.57 (d, *J* = 2.1 Hz, 2H), 5.55 (s, 2H). ^13^C NMR (100 MHz, *d*6-DMSO) *δ* 168.38, 161.34, 146.36, 145.13, 138.32, 129.06, 126.24, 123.19, 120.78, 111.05, 56.84. HRMS (ESI) *m/z* calcd. for C_16_H_11_F_3_NO_4_^−^ [M − H]^−^ 338.0646, found 338.0638.

*{5-Hydroxy-2-[2-(4-methanesulfonyl-phenyl)-vinyl]-4-oxo-4*H*-pyridin-1-yl}-acetic acid* (**7i**) (82%). Purity: 99.02%; m.p.: 275–277 °C; ^1^H NMR (400 MHz, *d*6-DMSO) *δ* 8.24 (s, 1H), 8.00 (d, *J* = 2.4 Hz, 4H), 7.61 (s, 1H), 7.56 (s, 2H), 5.49 (s, 2H), 3.26 (s, 3H). ^13^C NMR (100 MHz, *d*6-DMSO) *δ* 168.45, 161.87, 145.36, 141.63, 140.41, 137.88, 129.06, 127.96, 121.57, 111.15, 56.75, 43.93. HRMS (ESI) *m/z* calcd. for C_16_H_14_NO_6_S^−^ [M − H]^−^ 348.0547, found 348.0548.

*{2-[2-(4-Amino-phenyl)-vinyl]-5-hydroxy-4-oxo-4*H*-pyridin-1-yl}-acetic acid* (**7j**) (71%). Purity: 99.10%; m.p.: 222–224 °C; ^1^H NMR (400 MHz, *d*6-DMSO) *δ* 8.15 (s, 1H), 7.58 (s, 1H), 7.48 (d, *J* = 2.6 Hz, 2H), 7.32 (d, *J* = 8.7 Hz, 1H), 6.96 (d, *J* = 8.9 Hz, 1H), 6.70 (d, *J* = 2.2 Hz, 2H), 5.41 (s, 2H). ^13^C NMR (100 MHz, *d*6-DMSO) *δ* 169.02, 161.95, 150.37, 148.41, 144.59, 141.18, 132.13, 130.71, 124.56, 115.52, 112.06, 109.60, 56.78. HRMS (ESI) *m/z* calcd. for C_15_H_13_N_2_O_4_^−^ [M − H]^−^ 285.0881, found 285.0879.

*{5-Hydroxy-2-[2-(4-hydroxy-phenyl)-vinyl]-4-oxo-4*H*-pyridin-1-yl}-acetic acid* (**7k**) (32%). Purity: 95.14%; m.p.: 211–212 °C; ^1^H NMR (400 MHz, *d*6-DMSO) *δ* 12.19 (s, 1H), 9.80 (s, 1H), 8.40 (d, *J* = 2.1 Hz, 1H), 8.23 (s, 1H), 7.95 (d, *J* = 2.2 Hz, 1H), 7.71 (s, 1H), 7.49 (d, *J* = 8.7 Hz, 1H), 6.97 (m, 2H), 6.43 (s, 1H), 5.02 (s, 2H). ^13^C NMR (100 MHz, *d*6-DMSO) *δ* 169.03, 159.48, 148.79, 146.14, 143.77, 136.46, 128.93, 128.73, 117.01, 116.25. 106.60, 98.91, 56.24. HRMS (ESI) *m/z* calcd. for C_15_H_12_NO_5_^−^ [M − H]^−^ 286.0721, found 286.0708.

*{2-[2-(3,4-Dihydroxy-phenyl)-vinyl]-5-hydroxy-4-oxo-4*H*-pyridin-1-yl}-acetic acid* (**7l**) (29%). Purity: 96.21%; m.p. >300 °C; ^1^H NMR (400 MHz, *d*6-DMSO) *δ* 8.19 (s, 1H), 8.13 (s, 1H), 7.83 (d, *J* = 2.1 Hz, 1H), 7.79 (m, 2H), 7.68 (s, 1H), 6.93 (d, *J* = 8.1 Hz, 1H), 4.99 (s, 2H). ^13^C NMR (100 MHz, *d*6-DMSO) *δ* 178.80, 161.02, 152.94, 145.45, 135.27, 126.13, 125.75, 116.26, 116.14, 105.49, 104.03, 103.94, 56.88. HRMS (ESI) *m/z* calcd. for C_15_H_12_NO_6_^−^ [M − H]^−^ 302.0670, found 302.0687.

### 3.3. Enzyme Inhibition Assays

The ALR2 inhibition activity was evaluated in a mixture of reaction with 50 μL NADPH (0.1 mM), 50 μL sodium phosphate buffer (pH = 6.2, 100 mM), 50 μL human recombinant ALR2, 1 μL dimethyl sulfoxide (DMSO) as solvent, and 50 μL D,L-glyceraldehyde (10 mM) which served as a substrate. Although L-idose, which enable the transformation of ALR2, was an essential substrate for assays evaluating differential inhibitors, D,L-glyceraldehyde was used here to disclose classical inhibitors that interact as strongly as possible with the residues at the active site [41]. Before the substrate was added, the mixture was incubated at 30 °C for 3 min, then the reaction was started when the substrate was added, and was monitored for 4 min. The ALR1 inhibition activity experiment was conducted at 37 °C in the mixture containing 50 μL NADPH (0.12 mM), 50 *μ*L human recombinant ALR1, 50 μL sodium phosphate buffer (pH = 7.2, 100 mM), 50 μL sodium D-gluconate (20 mM) as a substrate, and 1 μL DMSO as solvent. Before the substrate sodium D-gluconate was added, the mixture was incubated at 37 °C for 3 min, then the substrate was added to start the reaction, which was monitored for 4 min.

The inhibitory activities of the title synthetic compounds against ALR2 and ALR1 were examined by adding 1 μL of the inhibitor solution to the ALR2 or ALR1 mixtures. All compounds were dissolved in DMSO, and the solutions were diluted with DMSO to desirable concentrations. To correct for the nonenzymatic oxidation of NADPH, the rate of NADPH oxidation in the presence of all of the reaction mixture components except the substrate was subtracted from each experimental rate. The title compounds were tested at concentrations ranging from 100 μM to 1 μM, except compound **7l** which were tested 10 μM to 0.1 μM. Epalrestat was tested at concentration from 1 μM to 0.01 μM. The inhibitory rates of tested compounds were calculated by equation as follows:Inhibitory rate (%) = (1 − ∆A_sample_/∆A_E+DMSO_) × 100%(1)
sample = enzyme + compound + DMSO(2)
E + DMSO = enzyme + DMSO(3)

As shown in Figure 3, most dose–response curves were generated using at least three concentrations of the compound with inhibitory rate, and the IC_50_ values were calculated by least-square analysis of the linear portion of the log(dose) versus response curves (r^2^ > 0.95), with equation IC_50_ (μM) = 10 ^x + 6^. The experiments were performed in triplicate.

### 3.4. Antioxidant Assays

Antioxidant assays were performed with the similar protocol as described previously [42].

### 3.5. Molecular Docking Assay

Molecular docking was performed utilizing Molegro Virtual Docker, version 6.0 with standard procedure [22,42]. The crystal structure of hALR2 conjugated with lidorestat (PDB code: 1Z3N) [38] and pure structure of hALR1 (PDB code: 1ZUA) [40] were downloaded from the RCSB Protein Data Bank to evaluate the binding mode of the derivatives. All solvent molecules within the protein document were deleted, and NADPH was set as cofactor. All 3D structural parameters of ligands were input into Molegro Virtual Docker software. The cavities were detected by Expanded Van der Waals around the enzyme surface, and cavity 1, which contained the anion binding site (volume of ~65 A), was selected for the origin of docking grid. The grid radius was set to 14. All the calculations were carried out by the grid-based MolDock score (GRID) function with a grid resolution of 0.20 A. The best ligand poses were chosen on the basis of ReRank score. The binding energy and inhibitory constant were calculated by the software. These docking assays were performed with a dual processor Windows 7 based computer with 4 GB RAM, and each docking process took 2–3 min.

## 4. Conclusions

To much effectively retard the long-term diabetic complications, antioxidative 5-hydroxy-2-styryl-1*H*-pyridin-4-ones-based derivatives were designed for a combination of selective ALR2 inhibition and antioxidant. Significantly, most of the synthetic **7** with styryl residues showed a good inhibitory activity, and some exhibited fine selectivity towards ALR2 and antioxidant activity. In addition, the results concluded from structure-activity relationship analysis combined with further molecular docking studies revealed that the substitution greatly altered inhibitory action of the derivatives. Substituted by alkine groups, hydroxyl groups and F atoms in the aromatic rings of the side chains, the inhibitory efficacy was enhanced dramatically; however, electronic-withdrawing groups, especially the methanesulfonyl group, decreased the potency. Particularly, molecular modeling study revealed that **7l** not only totally blocked the anion pocket but also strongly interacted with specificity pocket, proving that **7l,** of which IC_50_ (0.789 μM) was lowest among the derivatives, inhibited ALR2 much vigorously. More significantly, **7l** were found to be more active than conventional antioxidant Trolox, with DPPH radicals scavenging rate 41.48% (1 μM) and lipid peroxidation suppressing rate 88.76% (100 μM), suggesting that **7l** could favorably reduce the level of oxidative stress. In this study, antioxidant structure of 5-hydroxyl pyridinone was utilized to design novel multifunctional ALR2 inhibitors. As a result, compound **7l** represented potent leads for the achievement of the inhibitors possessing both capacities for ALR2 inhibition and as antioxidant.

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
