# Peer review of "(5-Hydroxy-4-oxo-2-styryl-4H-pyridin-1-yl)-acetic Acid Derivatives as Multifunctional Aldose Reductase Inhibitors"

_molecules, 2020, doi:10.3390/molecules25215135_

Round 1

Reviewer 1 Report

(5-Hydroxy-4-oxo-2-styryl-4H-pyridin-1-yl)-acetic-2-Acid Derivatives as Multifunctional Aldose 3 Reductase Inhibitors.

Abstracs

Describe that I performed the specificity test for aldolases R1 and R2, and why?

Information on molecular docking and antioxidant activity is missing, place concentration values.

It is suggested to rewrite, in the following order, explain the use of the brief aldolase, explain the results obtained first the synthesis of the compounds and then the selected activities, finally conclude.

Introduction

Information is lacking on 3-hydroxy-4 (1H) -pyridinone derivatives examples that have been used against ARL, and if they do not exist as antidiabetics in general. Review this references.

Taylor, P. D. (1996). Solution chemistry of vanadium-(IV) and-(V) with the bidentate ligand 1, 2-dimethyl-3-hydroxy-4 (1H)-pyridinone: relevance to the treatment of diabetes. Chemical Communications, (3), 405-406.

Katoh, A., Yamaguchi, M., Taguchi, K., Saito, R., Adachi, Y., Yoshikawa, Y., & Sakurai, H. (2006). Oxovanadium (IV) and (V) Complexes with 3-Hydroxy-4 (1H)-pyridinones and 1-Hydroxy-2 (1H)-pyrimidinones—Synthesis, Structural Characteristics, and Their Insulin-mimetic Activities. Biomedical research on trace elements, 17(1), 1-10.

Rangel, M., Tamura, A., Fukushima, C., & Sakurai, H. (2001). In vitro study of the insulin-like action of vanadyl-pyrone and-pyridinone complexes with a VO (O 4) coordination mode. JBIC Journal of Biological Inorganic Chemistry, 6(2), 128-132.

Results and Discussion

In the chemical synthesis there is no discussion, they only mention the results obtained.

The enzymatic activity presented is not compared with the literature.

The DPPH activity presented is not compared to the literature.

Lipid peroxidation suppression is not compared to the literature.

The results of Molecular docking were not compared with the literature.

Figure 3 does not clearly show the molecular coupling, I suggest changing the black background for a white one and highlighting with a striking color the amino acids that interfere with the active center found. It is necessary to look for a suitable figure to demonstrate the interactions, the one presented in this work is complicated for the reader to observe the interactions and the amino acid residues involved, you could use a 2D visualization using the free Discovery Studio program. 

From the coupling energy data, Ki could be calculated and related to the inhibition values found in the experimental part.

Materials and Methods

13C NMR spectra are missing for all reported intermediates. (If they are already reported in the literature, please give the references).

Molecular Docking. In this case, there is already a ligand that can be removed as an active site, why don't I use this for the experiment? What are the coordinates used? What is the size of the grid? How do I perform active site validation?

Conclusions.

The work has many interesting results, the conclusion does not represent the work done by the authors, it is very weak.

Author Response

Reviewer 1’s comment:

Abstracs

  1. Describe that I performed the specificity test for aldolases R1 and R2, and why?; Information on molecular docking and antioxidant activity is missing, place concentration values.; It is suggested to rewrite, in the following order, explain the use of the brief aldolase, explain the results obtained first the synthesis of the compounds and then the selected activities, finally conclude.

 Respond:

Abstract was rewritten as suggested in following order: introduction target enzyme and multifunctional ARI, design of the compounds, main results and final conclusion. (page 1) The reason why performing specificity of ALR1 and ALR2 was mentioned in line 11 in the revised version. Detail information of molecular docking and antioxidant activity were included in the abstract of the revised version. 

Introduction

  1. Information is lacking on 3-hydroxy-4 (1H) -pyridinone derivatives examples that have been used against ARL, and if they do not exist as antidiabetics in general. Review this references.

Taylor, P. D. (1996). Solution chemistry of vanadium-(IV) and-(V) with the bidentate ligand 1, 2-dimethyl-3-hydroxy-4 (1H)-pyridinone: relevance to the treatment of diabetes. Chemical Communications, (3), 405-406.

Katoh, A., Yamaguchi, M., Taguchi, K., Saito, R., Adachi, Y., Yoshikawa, Y., & Sakurai, H. (2006). Oxovanadium (IV) and (V) Complexes with 3-Hydroxy-4 (1H)-pyridinones and 1-Hydroxy-2 (1H)-pyrimidinones—Synthesis, Structural Characteristics, and Their Insulin-mimetic Activities. Biomedical research on trace elements, 17(1), 1-10.

Rangel, M., Tamura, A., Fukushima, C., & Sakurai, H. (2001). In vitro study of the insulin-like action of vanadyl-pyrone and-pyridinone complexes with a VO (O 4) coordination mode. JBIC Journal of Biological Inorganic Chemistry, 6(2), 128-132.

 Respond:

Thank you for providing information about the application of 3-hydroxy-4 (1H) -pyridinone derivatives. Indeed, as reviewed in the introduction in original manuscript, 3-hydroxy-4 (1H) -pyridinone derivatives were developed as antidiabetic agents. However, there is not any derivative that have been applied in ARI (mentioned in revised version, Line 85, Page 3). Thus, for potential antidiabetic complication therapy strengthened by ALR2 inhibition, the styryl type carbonxylic derivatives were chosen to ARI design with antioxidant property.

Besides, to improve the quality of the reviewing, the listed references were cited in general review of the derivatives. (Line 77, in page 2)

Results and Discussion

  1. In the chemical synthesis there is no discussion, they only mention the results obtained.

Respond:

In revised version, chemical synthesis was briefly discussed by means of optimizing reaction condition and spectroscopy. (line 104, page 3)

  1. The enzymatic activity presented is not compared with the literature.

Respond:

Thank you for your kind advise to improve the discussion section informatively. To ensure the validity of the test, eparlestat, the positive control was compared with the literature that recently published and with similar method. (Line 137, page 4)

  1. The DPPH activity presented is not compared to the literature.; Lipid peroxidation suppression is not compared to the literature.

Respond:

The antioxidant activities of tested compounds were compared to reported antioxidant deferiprone (3-hydroxy-1,2-dimethyl-1H-pyridin-4-one), which shared similar structure of the compound listed in this paper. (line 163, page 6, for DPPH assays; line 189, page 7, for lipid peroxidation assays)

  1. The results of Molecular docking were not compared with the literature.

Respond:

Molecular docking results, especially interactions between inhibitors and ALR2, were compared to cemtirestat O-analog reported recently. (line 217, page 8)

  1. Figure 3 does not clearly show the molecular coupling, I suggest changing the black background for a white one and highlighting with a striking color the amino acids that interfere with the active center found. It is necessary to look for a suitable figure to demonstrate the interactions, the one presented in this work is complicated for the reader to observe the interactions and the amino acid residues involved, you could use a 2D visualization using the free Discovery Studio program. 

Respond:

Thank you for your kind advice, easily readable docking figure was remade by using 2D visualization with software Discovery Studio. (fig 2, page 9)

  1. From the coupling energy data, Ki could be calculated and related to the inhibition values found in the experimental part.

 Respond:

Ki was calculated (table 4,page 10) and discussed with the IC50 data derived from experiment section. (line 228, page 8)

Materials and Methods

  1. 13C NMR spectra are missing for all reported intermediates. (If they are already reported in the literature, please give the references).

Respond:

Compounds 2-4 were reported by Xu, et al.(Xu, P.; Zhang, M.; Sheng, R.; Ma, Y. Synthesis and biological evaluation of deferiprone-resveratrol hybrids as antioxidants, Aβ1–42 aggregation inhibitors and metal-chelating agents for Alzheimer's disease. Eur. J. Med. Chem. 2017, 127, 174.).13C NMR of other compounds were given in section of method.(page 11)

  1. Molecular Docking. In this case, there is already a ligand that can be removed as an active site, why don't I use this for the experiment? What are the coordinates used? What is the size of the grid? How do I perform active site validation?

 Respond:

In molecular docking assay, reported standard procedure stimulating binding mode of inhibitors was processed (Xin Hao, Gang Qi, Hongxing Ma, Changjin Zhu & Zhongfei Han (2019) Novel 2-phenoxypyrido[3,2-b]pyrazin-3(4H)-one derivatives as potent and selective aldose reductase inhibitors with antioxidant activity, Journal of Enzyme Inhibition and Medicinal Chemistry, 34:1, 1368-1372). In template 1z3n, there are a ligand lidorestat and a cofactor NADPH. The active site of the enzyme is the anion binding pocket, which is mainly consisted by residues His 110, Trp 111, Tyr 48, Val 47, Trp 20 and cofactor NADPH. In this case, ligand lidorestat which is not the hydroxyl-pyridinones, was not used in this section which aimed at evaluation of the binding mode of title compounds. 

Preparation of the docking: 3D molecules of the tested compounds were input into the software MVD. The cavities were calculated by means of enzyme surface detection with Expanded Van der Waal. The cavity 1 which containing the anion binding site (volume of ~65 A) was selected for the origin of docking grid. The grid radius was set to 14.

Conclusions.

11.The work has many interesting results, the conclusion does not represent the work done by the authors, it is very weak.

Respond:

Conclusion was rewritten according to the design of the pyridinones as multifunctional inhibitor. Main results of ALR2 inhibition and antioxidant were emphasized to prove that 5-hydroxyl pyridinone compound 7l was qualified multifunctional inhibitor. (page 10)

Reviewer 2 Report

Manuscript title: (5-Hydroxy-4-oxo-2-styryl-4H-pyridin-1-yl)-acetic Acid Derivatives as Multifunctional Aldose Reductase Inhibitors

Aldose reductase is an important enzyme catalyzing reduction of glucose to sorbitol by polyol pathway. This pathway leads to several complications in diabetes peoples v.z. microvascular damage many organs. Thus, aldose reductase inhibitors can prevent diabetic complications. This study reports the synthesis of pyridinone derivatives and their in vitro aldose reductase inhibitory activity. The results are attempted to be rationalized by molecular docking studies. The authors are urged to consider the following comments;

  1. Delete the methodology part from the abstract and include only major findings.

  1. The introduction contains little justification for the work presented in this MS. Please include a short review of similar experimental/computational studies involving aldose reductase inhibitors. Additionally, the introduction can be strengthened by clarifying the novelty of the study.

  1. It would be informative to include the NMR/UV-Vis spectra of the synthesized compounds into Supporting Information.

  1. Why have the authors chosen 1Z3N and 3H4G PDB structures? 1Z3N has pretty low resolution (1.04 Angstrom) and is quite old, while 3H4G is of the pig. The human aldose reductase is available e.g., 2R24 has a resolution of ~1.7 Angstrom and is co-crystallized with the small molecule ligand IDD594. This needs clarification with high impact journal citations, or the author would consider using 2R24 PDB.

  1. In the main text, the authors mention that IC50 was determined using 100 μM to 10 nM Kindly provide raw data in the supplementary information.

  1. A few exemplary dose-response graphs should be included to show the determination of IC50.

  1. It would be informative to employ a statistical method such as ANOVA with Tukey test to analyze the statistical significance between both inhibition and IC50 between the synthesized compounds.

  1. In the Conclusions, the authors state that the C2 dihydroxystyryl side chain was proved as the critical structure in the 5-hydroxyl pyridinones compounds 7l representing potent leads for the achievement of multifunctional ARIs. This is quite a broad claim. It would be very informative for the readers to include more specific design considerations that are a result of this study.

  1. Delete from the conclusion, “In this paper, a series of 5-hydroxy-2-styryl-1H-pyridin-4-ones based derivatives were designed and synthesized by changing of the substituent in C2 styryl side chain to refine these ALR2 inhibitors”.

Author Response

Reviwer 2’s comment:

Aldose reductase is an important enzyme catalyzing reduction of glucose to sorbitol by polyol pathway. This pathway leads to several complications in diabetes peoples v.z. microvascular damage many organs. Thus, aldose reductase inhibitors can prevent diabetic complications. This study reports the synthesis of pyridinone derivatives and their in vitro aldose reductase inhibitory activity. The results are attempted to be rationalized by molecular docking studies. The authors are urged to consider the following comments;

  1. Delete the methodology part from the abstract and include only major findings.

 Respond:

Abstract was totally rewritten including major results and findings, such as inhibitory activity, selectivity, antioxidant efficacy and molecular docking, and excluding methodology.(page 1)

  1. The introduction contains little justification for the work presented in this MS. Please include a short review of similar experimental/computational studies involving aldose reductase inhibitors. Additionally, the introduction can be strengthened by clarifying the novelty of the study.

 Respond:

Aldose reductase inhibitors developed recently were reviewed briefly in the introduction section. Endowed with potent ALR2 inhibitory activity, compounds with carboxylic acid moiety were mainly developed recently among synthetic ARIs. 2-(3-Oxo-2H-[1,2,4]triazino[5,6-b]indol-5(3H)-yl) acetic acid, 2-phenoxypyrido[3,2-b]pyrazin-3(4H)-one, (4-oxo-2-thioxothiazolidin-3-yl) acetic acid and rhodanine-3-hippuric acid derivatives showed high inhibitory efficacy with attractive IC50 values (23 ~ 42 nM), suggesting that carboxylic group is the key moiety preforming outstanding activity. However, the antioxidant properties of these inhibitors need to be improved. (Line 70, Page 2). 

Meanwhile, the novelty of this work was strengthened by clarifying the design of carbonxylic derivatives with potent antioxidantive core. “Thus, combining ALR2 inhibition with antioxidant action, hydroxyl pyridinones is favorable to be designed as antioxidative core of carbonxylic multifunctional ARI.” (Line 83, Page 3)

  1. It would be informative to include the NMR/UV-Vis spectra of the synthesized compounds into Supporting Information.

 Respond:

NMR and UV spectra of title compounds were included in Supporting Information of revised version (page 1 and page 16, respectively).

  1. Why have the authors chosen 1Z3N and 3H4G PDB structures? 1Z3N has pretty low resolution (1.04 Angstrom) and is quite old, while 3H4G is of the pig. The human aldose reductase is available e.g., 2R24 has a resolution of ~1.7 Angstrom and is co-crystallized with the small molecule ligand IDD594. This needs clarification with high impact journal citations, or the author would consider using 2R24 PDB.

 Respond:

Because chemical structures of title compounds are similar to that of lidorestat, especially sharing similar spaces and angles between carbonxylic groups and sidechains, 1Z3N is appropriate for docking stimulation of title compounds. 

Thanks for your advise using a template with higher resolution to improve docking experiment. However, according to PDB website, it is called high-resolution structure with a low resolution value. (http://pdb101.rcsb.org/learn/guide-to-understanding-pdb-data/resolution) 
Thus, 1Z3N has higher resolution than that 2R24 does. 

Indeed, 3H4G which is of Sus scrofa, was improper to be used as the template in original manuscript. In revised version, as you suggested by using template with higher solution, we decide to change 3H4G into 1ZUA which is of attractive solution and derived from human AKR1B10 crystal structure. The rerank scores was calculated with ALR1/NADP+ complex (PDB: 1ZUA) by MVD software. (Line 234 page 8)

  1. In the main text, the authors mention that IC50 was determined using 100 μM to 10 nM Kindly provide raw data in the supplementary information.

 Respond:

Sorry for being misled and confused by the bad editing of method description in original manuscript. In ALR2 inhibitory tests, final concentrations of all title compounds were 100 μM to 1 μM, except that of 7l, which were 10 μM to 0.1 μM. Eparlestat, serving as positive control, were tested in concentrations of 1 μM to 0.01 μM. In experiment section of revised version, the description was carefully reedited as clearly as possible (Line 498, Page 16). The raw data is provided in Supporting information of revised version.

  1. A few exemplary dose-response graphs should be included to show the determination of IC50.

 Respond:

Exemplary log(dose) vs inhibitory rate graph was provided in experiment section of revised version. (Fig 3, page 16)

  1. It would be informative to employ a statistical method such as ANOVA with Tukey test to analyze the statistical significance between both inhibition and IC50 between the synthesized compounds.

Respond:

Thank you very much for your excellent advise to improve the discussion about the inhibitory activity. ALR2 IC50 values were analyzed by one-way ANOVA Tukey tests (table 1, page 5) to verify the significance of the inhibitory potency between the title compounds with different substitutions in aromatic rings of the side chain (Line 126, 135, page 3). Besides, the significant effect of phenolic hydroxyl group was concluded by the significancy analysis between 7d and 7k. (line 131, page 3)

  1. In the Conclusions, the authors state that the C2 dihydroxystyryl side chain was proved as the critical structure in the 5-hydroxyl pyridinones compounds 7l representing potent leads for the achievement of multifunctional ARIs. This is quite a broad claim. It would be very informative for the readers to include more specific design considerations that are a result of this study.

 Respond:

Conclusion was rewritten according to the design of the pyridinones as multifunctional inhibitor. Main results of ALR2 inhibition and antioxidant were emphasized to prove that 5-hydroxyl pyridinone compound 7l was qualified multifunctional inhibitor. (page 10)

  1. Delete from the conclusion, “In this paper, a series of 5-hydroxy-2-styryl-1H-pyridin-4-ones based derivatives were designed and synthesized by changing of the substituent in C2 styryl side chain to refine these ALR2 inhibitors”.

Respond:

The first sentence in conclusion of origin manuscript, “In this paper, a series of 5-hydroxy-2-styryl-1H-pyridin-4-ones based derivatives were designed and synthesized by changing of the substituent in C2 styryl side chain to refine these ALR2 inhibitors”was deleted in the revised version.

Reviewer 3 Report

In the Manuscript ID: molecules-972985, the authors report design and synthesis of a series of pyridinone derivatives, and evaluated their inhibitory behavior by in vitro aldose reductase inhibition and aldehyde reductase inhibition experiments, and antioxidant activity by DPPH radical and MDA suppressing assays.

Althogh it seems to be tough for preparation of glycine substitution derivative from pyranone, the paper is well-written, the tables and figures are of high quality, and the authors have clearly detailed description of their methods.

I am interesting in the ALR inhibition activiry for pyranone derivatives 5a-j, because it will be clear the importancy of carboxylic acid unit.

The following style comments should be considered by the authors;

#1 Line 193 ...these data were...

#2 Each pictures and structures in Figure 3 are too small.

#3 Please refer some recent reports about aldose reductase inhibitors including acetic acid moiety shown below ;

Kucerova-Chlupacova Marta; Halakova Dominika; Majekova Magdalena; Stefek Milan; Soltesova Prnova Marta; Treml Jakub

(4-​oxo-​2-​thioxothiazolidin-​3-​yl)​acetic acids as potent and selective aldose reductase inhibitors

Chemico-biological interactions (2020), 109286.

https://doi.org/10.1016/j.cbi.2020.109286

Celestina Stephen Kumar; Ravi Subban; Sundaram Kaveri

In vitro studies of potent aldose reductase inhibitors: Synthesis, characterization, biological evaluation and docking analysis of rhodanine-​3-​hippuric acid derivatives

Bioorganic chemistry (2020), 97, 103640.

https://doi.org/10.1016/j.bioorg.2020.103640

Hao Xin; Hao Xin; Zhu Changjin; Han Zhongfei; Qi Gang; Ma Hongxing; Han Zhongfei

Novel 2-​phenoxypyrido[3,​2-​b]​pyrazin-​3(4H)​-​one derivatives as potent and selective aldose reductase inhibitors with antioxidant activity

Journal of enzyme inhibition and medicinal chemistry (2019), 34(1), 1368-1372.

https://doi.org/10.1080/14756366.2019.1643336

Soltesova Prnova Marta; Svik Karol; Bezek Stefan; Kovacikova Lucia; Stefek Milan; Karasu Cimen

3-​Mercapto-​5H-​1,​2,​4-​Triazino[5,​6-​b]​Indole-​5-​Acetic Acid (Cemtirestat) Alleviates Symptoms of Peripheral Diabetic Neuropathy in Zucker Diabetic Fatty (ZDF) Rats: A Role of Aldose Reductase

Neurochemical research (2019), 44(5), 1056-1064.

https://doi.org/10.1007/s11064-019-02736-1

Author Response

Reviewer 3’s comment:

Althogh it seems to be tough for preparation of glycine substitution derivative from pyranone, the paper is well-written, the tables and figures are of high quality, and the authors have clearly detailed description of their methods.

I am interesting in the ALR inhibition activiry for pyranone derivatives 5a-j, because it will be clear the importancy of carboxylic acid unit.

Respond:

We have had a glace at the inhibitory action of 5d, which is inactive (I% < 20%) in testing concentration of 10 μM. It suggested that carboxylic group is important for the inhibitor.

The following style comments should be considered by the authors;

#1 Line 193 ...these data were...

Respond: 

Thanks for your carefulness. The sentence was revised to “...these data were explicit that...” (line 236, page 8)

#2 Each pictures and structures in Figure 3 are too small.

Respond:

Thank you for your kind advice, easily readable docking figure was remade by using 2D visualization with software Discovery Studio. (fig 2, page 9)

#3 Please refer some recent reports about aldose reductase inhibitors including acetic acid moiety shown below ;

Kucerova-Chlupacova Marta; Halakova Dominika; Majekova Magdalena; Stefek Milan; Soltesova Prnova Marta; Treml Jakub

(4-​oxo-​2-​thioxothiazolidin-​3-​yl)​acetic acids as potent and selective aldose reductase inhibitors

Chemico-biological interactions (2020), 109286.

https://doi.org/10.1016/j.cbi.2020.109286

Celestina Stephen Kumar; Ravi Subban; Sundaram Kaveri

In vitro studies of potent aldose reductase inhibitors: Synthesis, characterization, biological evaluation and docking analysis of rhodanine-​3-​hippuric acid derivatives

Bioorganic chemistry (2020), 97, 103640.

https://doi.org/10.1016/j.bioorg.2020.103640

Hao Xin; Hao Xin; Zhu Changjin; Han Zhongfei; Qi Gang; Ma Hongxing; Han Zhongfei

Novel 2-​phenoxypyrido[3,​2-​b]​pyrazin-​3(4H)​-​one derivatives as potent and selective aldose reductase inhibitors with antioxidant activity

Journal of enzyme inhibition and medicinal chemistry (2019), 34(1), 1368-1372.

https://doi.org/10.1080/14756366.2019.1643336

Soltesova Prnova Marta; Svik Karol; Bezek Stefan; Kovacikova Lucia; Stefek Milan; Karasu Cimen

3-​Mercapto-​5H-​1,​2,​4-​Triazino[5,​6-​b]​Indole-​5-​Acetic Acid (Cemtirestat) Alleviates Symptoms of Peripheral Diabetic Neuropathy in Zucker Diabetic Fatty (ZDF) Rats: A Role of Aldose Reductase

Neurochemical research (2019), 44(5), 1056-1064.

https://doi.org/10.1007/s11064-019-02736-1

Respond:

Thank you for your kindy providing reference. Recently reported aldose reductase inhibitors were reviewed briefly in introduction section. Some of the listed references were cited.

Endowed with potent ALR2 inhibitory activity, compounds with carboxylic acid moiety were mainly developed recently among synthetic ARIs. 2-(3-Oxo-2H-[1,2,4]triazino[5,6-b]indol-5(3H)-yl) acetic acid, 2-phenoxypyrido[3,2-b]pyrazin-3(4H)-one, (4-oxo-2-thioxothiazolidin-3-yl) acetic acid and rhodanine-3-hippuric acid derivatives showed high inhibitory efficacy with attractive IC50 values (23 ~ 42 nM), suggesting that carboxylic group is the key moiety preforming outstanding activity. However, the antioxidant properties of these inhibitors need to be improved. (Line 70, Page 2). 

Reviewer 4 Report

The Authors describe the synthesis and characterization of a series of compounds to be used as multifunctional aldose reductase inhibitors. Even though data presented could be interesting, the manuscript needs major revisions to be considered for publication, especially in the part describing the enzyme inhibition studies.

General Comment.

The manuscript is not always clear and English is poor in some points. A review of the manuscript by a native English speaker is highly recommended.

Major points to be addressed.

The Authors use, for the kinetic characterization of their inhibitors, enzyme preparations of aldose reductase and aldehyde reductase. The Authors does not give any information about their enzymes. Are aldose reductase and aldehyde reductase pure enzymes? How were the enzyme preparations obtained? While the use of partially purified enzymes was acceptable some years ago, today that is no more acceptable, since several recombinant and purification techniques are available for obtaining pure enzymes. If the Authors used partially purified enzymes, have they verified that their preparations of aldose reductase and aldehyde reductase did not contain any activity that could interfere with the enzymatic assay? What checks did they do?

The choice of the substrate used (glyceraldehyde) for aldose reductase assay is not completely adequate. It is true that glyceraldehyde is a classical substrate used for aldose reductase, but in recent years some articles have pointed out that the reduction of glucose by aldose reductase occurs under different conditions with respect to glyceraldehyde, due to the presence of the hemiacetal form of glucose. Thus, in studies in which molecules are searched for the inhibition of the reduction of glucose by aldose reductase, glucose is the appropriate substrate. Glucose is a very poor substrate for aldose reductase, but alternative substrates (e.g. L-idose) have been proposed for aldose reductase assay. The Authors should comment on their choice of the substrate for aldose reductase, acknowledging recent literature on the subject.

All compounds used were dissolved in DMSO. Since DMSO has been reported to be an aldose reductase inhibitor, did the Authors checked the inhibition of aldose reductase by DMSO under their experimental conditions?

In the Introduction, the Authors discuss the question about why so far aldose reductase inhibitors have failed as molecules for the treatment of diabetic complications and propose their molecules as multifunctional aldose reductase inhibitors. However, they completely neglect (disregard) articles appeared in recent literature on the same argument, that propose the use of molecules able to inhibit the reduction of glucose by aldose reductase, without blocking the capability of aldose reductase itself to reduce toxic aldehydes.

Minor points

Figure 1. Glucose 6-P can be used in glycolysis and in pentose phosphate pathway; thus indicating only pentose phosphate pathway is misleading.

In the description of aldose reductase and aldehyde reductase assays the same units for volumes (mL or µL) and for concentrations (mM or µM) should be used.

The Authors state that IC50 values were calculated from log (dose) versus response curves; how can they give IC50 values + standard error? If a log (dose) versus response curve is used, one should give the confidence interval.

Author Response

Reviewer 4’s comment:

General Comment.

The manuscript is not always clear and English is poor in some points. A review of the manuscript by a native English speaker is highly recommended.

Respond:

We have checked and improved the article with standard English.

Major points to be addressed.

  1. The Authors use, for the kinetic characterization of their inhibitors, enzyme preparations of aldose reductase and aldehyde reductase. The Authors does not give any information about their enzymes. Are aldose reductase and aldehyde reductase pure enzymes? How were the enzyme preparations obtained? While the use of partially purified enzymes was acceptable some years ago, today that is no more acceptable, since several recombinant and purification techniques are available for obtaining pure enzymes. If the Authors used partially purified enzymes, have they verified that their preparations of aldose reductase and aldehyde reductase did not contain any activity that could interfere with the enzymatic assay? What checks did they do?

Respond:

Thank you for your concerns and suggestions to improve our manuscript. Aldose reductase and aldehyde reductase were prepared and partially purified in original manuscript. In revised version, the human recombinant AKR1B1 and AKR1B10 purchased from sigma-aldrich were used as the enzymes. The results and discussion section were rewritten (line 119, page 4), as well as method description (line 481 page 15). The main conclusion derived from the enzyme tests were similar to that of original manuscript.

  1. The choice of the substrate used (glyceraldehyde) for aldose reductase assay is not completely adequate. It is true that glyceraldehyde is a classical substrate used for aldose reductase, but in recent years some articles have pointed out that the reduction of glucose by aldose reductase occurs under different conditions with respect to glyceraldehyde, due to the presence of the hemiacetal form of glucose. Thus, in studies in which molecules are searched for the inhibition of the reduction of glucose by aldose reductase, glucose is the appropriate substrate. Glucose is a very poor substrate for aldose reductase, but alternative substrates (e.g. L-idose) have been proposed for aldose reductase assay. The Authors should comment on their choice of the substrate for aldose reductase, acknowledging recent literature on the subject.

Respond:

Aimed at the improvement of both selectivity towards the target enzyme and inhibition potency and supported by molecular docking, in this paper, the strategic approach to developing hydroxyl-pyridinones as classical inhibitors was adopted. Not of a differential inhibitor, this strategy is typically to disclose molecules able to specifically intervene on the target enzyme, and that interact as strongly as possible with the residues at the active site. Thus, hemiacetal substrate (e.g. L-idose) which is essential for DIs evaluation to enable the transformation of the enzyme [Balestri, F.; Cappiello, M.; Moschini, R.; Rotondo, R.; Abate, M.; Del-Corso, A.; Mura, U. Modulation of aldose reductase activity by aldose hemiacetals. BBA-Gen. Subjects. 2015, 1850, 2329.], is not necessary for classical inhibitors [Cappiello, M.; Balestri, F.; Moschini, R.; Mura, U.; Del-Corso, A. Intra-site differential inhibition of multi-specific enzymes. J. Enzym. Inhib. Med. Ch. 2020, 35, 840-846.]. Indeed, especially in these two years, most of classical inhibitors were estimated by the standard method using glyceraldehyde [Kucerova-Chlupacova, M.; Halakova, D.; Majekova, M.; Treml, J.; Stefek, M.; Prnova, M. S. (4-Oxo-2-thioxothiazolidin-3-yl)acetic acids as potent and selective aldose reductase inhibitors. Chem-Biol Interact. 2020332, 109286. ; Celestina, S. K.; Sundaram, K.; Ravi, S. In vitro studies of potent aldose reductase inhibitors: Synthesis, characterization, biological evaluation and docking analysis of rhodanine-3-hippuric acid derivatives. Bioorg. Chem. 202097, 103640. ; Hao, X.; Qi, G.; Ma, H.; Zhu, C.; Han Z. Novel 2-phenoxypyrido[3,2-b]pyrazin-3(4H)-one derivatives as potent and selective aldose reductase inhibitors with antioxidant activity. J. Enzym. Inhib. Med. Ch. 201934, 1368–1372.; Hlavá c, M.; Ková č ikova, L; Prnova, M. S.; Sramel, P.; Addova, G.; Majekova, M.; Hanquet, G.; Bohá c, A.; Stefek, M. Development of Novel Oxotriazinoindole Inhibitors of Aldose Reductase: Isosteric Sulfur/Oxygen Replacement in the Thioxotriazinoindole Cemtirestat Markedly Improved Inhibition Selectivity. J. Med. Chem. 202063, 369−381.], proving that this classical substrate is acceptable. In conclusion, glyceraldehyde was qualified to be chosen when estimating ALR2 activity.

The choice of D,L-glyceraldehyde was stated in the section of method.(line 483, page 15)

  1. All compounds used were dissolved in DMSO. Since DMSO has been reported to be an aldose reductase inhibitor, did the Authors checked the inhibition of aldose reductase by DMSO under their experimental conditions?

Respond:

It was reported that DMSO is a weak inhibitor against ALR2. Indeed, when recently using recombination AKR1B1, 0.5% DMSO, which was identical concentration in the tests, inhibit ALR2 by inhibitory rate of 25%. Thus, the inhibitory rates of tested compounds were calculated by equation as followed: 

Inhibitory rate (%) = (1 - ∆Asample/∆AE+DMSO) × 100%

Here: sample = enzyme + compound + DMSO; E+DMSO = enzyme + DMSO.

(line 500, page 16)

  1. In the Introduction, the Authors discuss the question about why so far aldose reductase inhibitors have failed as molecules for the treatment of diabetic complications and propose their molecules as multifunctional aldose reductase inhibitors. However, they completely neglect (disregard) articles appeared in recent literature on the same argument, that propose the use of molecules able to inhibit the reduction of glucose by aldose reductase, without blocking the capability of aldose reductase itself to reduce toxic aldehydes.

 Respond:

In the introduction, area of differential inhibitor was mentioned in revised version (line 62 page 2), despite that developing multifunctional classical inhibitor with ALR2 selectivity and antioxidant effect was focused in this manuscript.

Minor points

  1. Figure 1. Glucose 6-P can be used in glycolysis and in pentose phosphate pathway; thus indicating only pentose phosphate pathway is misleading.

Respond:

Figure 1 was remade to emphasize the polyol pathway. (page 2)

  1. In the description of aldose reductase and aldehyde reductase assays the same units for volumes (mL or µL) and for concentrations (mM or µM) should be used.

Respond:

In the description about ingredient of reaction mixture, units for volumes and for concentration was unified as µL and mM respectively. (line 480, page 15)

  1. The Authors state that IC50 values were calculated from log (dose) versus response curves; how can they give IC50 values + standard error? If a log (dose) versus response curve is used, one should give the confidence interval.

Respond:

To minimize the environmental error, IC50 values are the mean±SD of triplicate measurements, which is standard procedure [Kucerova-Chlupacova, M.; Halakova, D.; Majekova, M.; Treml, J.; Stefek, M.; Prnova, M. S. (4-Oxo-2-thioxothiazolidin-3-yl)acetic acids as potent and selective aldose reductase inhibitors. Chem-Biol Interact. 2020332, 109286. ; Celestina, S. K.; Sundaram, K.; Ravi, S. In vitro studies of potent aldose reductase inhibitors: Synthesis, characterization, biological evaluation and docking analysis of rhodanine-3-hippuric acid derivatives. Bioorg. Chem. 202097, 103640. ; Hao, X.; Qi, G.; Ma, H.; Zhu, C.; Han Z. Novel 2-phenoxypyrido[3,2-b]pyrazin-3(4H)-one derivatives as potent and selective aldose reductase inhibitors with antioxidant activity. J. Enzym. Inhib. Med. Ch. 201934, 1368–1372.; Hlavá c, M.; Ková č ikova, L; Prnova, M. S.; Sramel, P.; Addova, G.; Majekova, M.; Hanquet, G.; Bohá c, A.; Stefek, M. Development of Novel Oxotriazinoindole Inhibitors of Aldose Reductase: Isosteric Sulfur/Oxygen Replacement in the Thioxotriazinoindole Cemtirestat Markedly Improved Inhibition Selectivity. J. Med. Chem. 202063, 369−381.]. (line 508, page 16)

Round 2

Reviewer 1 Report

The work presented has improved a lot. There are no more recommendations.

Reviewer 2 Report

The authors have satisfactorily answered my comments. The manuscript may be accepted for publication in Molecules in the present form now.

Reviewer 4 Report

The Authors fulfilled all the points raised.